# Working Capital Management and Shareholder's Wealth Creation: Evidence from Manufacturing Companies Listed in Oman

Shrikant Krupasindhu Panigrahi [1,*], Maryam Juma Al Farsi [2], Sumathi Kumaraswamy [1], Muhammad Waris Ali Khan [3,*] and Faisal Rana [4]

[1] Department of Economics and Finance, College of Business Administration, University of Bahrain, Sakhir P.O. Box 32038, Bahrain
[2] College of Business, University of Buraimi, Al Buraimi P.O. Box 562, Oman
[3] Faculty of Business and Law, The British University in Dubai, Dubai P.O. Box 345015, United Arab Emirates
[4] School of Business Administration, American University in Dubai, Dubai P.O. Box 28282, United Arab Emirates
* Correspondence: spanigrahi@uob.edu.bh (S.K.P.); waris.khan@buid.ac.ae (M.W.A.K.)

**Abstract:** Working capital management (WCM) is a key factor in the success of manufacturing companies when credit is restricted, as is the case in the current climate caused by the COVID-19 crisis. The main purpose of this paper is to investigate the relationship between working capital management, earnings quality, sales growth, and shareholders' wealth of listed manufacturing firms in Oman. The study used balanced panel data of 31 manufacturing firms listed on the Muscat Stock Exchange (MSE) from 2004 to 2019. The study reveals that days in working capital, cash conversion cycle, payable deferred period, sales growth, and earnings quality positively affects shareholder's wealth proxied by the return on assets, whereas, days in working capital have a negative effect on return on assets. Similarly, working capital management was found to have no influence on the earnings per share (EPS). It was also documented that sales growth and earnings quality positively impacted EPS. The study concluded that improving sales growth and earnings quality would result in shareholders' wealth creation. The results are helpful to manufacturing companies to improve their business performance and social welfare through a direct and indirect chain of raising investments, pay, and production scales. This study adds knowledge to the body of literature on working capital management, earnings quality, and sales growth in the areas of methodology, the impact of WCM components on manufacturing firms' shareholder value, and socioeconomic evidence from Oman.

**Keywords:** corporate finance; working capital; earnings quality; shareholder's wealth; sales growth; panel analysis

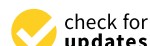



## 1. Introduction

In the wake of the COVID-19 pandemic crisis, the debate over how shareholder's wealth can be preserved has received growing attention in recent years. The pandemic led to a sharp increase in corporate's attention to survive and preserving shareholder's value. In every aspect of contemporary life, the pandemic with the coronavirus illness disrupted the economy by unbalancing consumption and production (Pourmansouri et al. 2022). The virus not only disrupted the Oman economy, but also the corporate and shareholder's wealth were eroded significantly. Organizations were forced to make changes in their business mechanisms and working capital strategies to improve their quality of earnings after COVID disruption.

Earnings quality refers to the reliability of the corporate earnings that assess corporate future performance (Jeong and Choi 2019). Earnings quality plays an important role for any company to show its strong financial well-being and capital efficiency. Investors and

financial analysts monitor a firm's earnings closely as a leading source of information on future cash flows and financial performance (Ayu et al. 2020). WCM refers to the management of current liabilities and current assets. Working capital management (WCM) is a key factor in the success of manufacturing companies when credit is restricted, as is the case in the current climate caused by the COVID-19 crisis. For a successful business, it is very important to optimize each element of WCM that influences financial performance (Filbeck and Krueger 2005; Sawarni et al. 2020). Conversely, business continuation would also be affected leading to profitability decline or losing market share if the working capital is not managed appropriately (Bhatia and Srivastava 2016; Seth et al. 2020b). WCM is an important element and requires continuous monitoring of the firm's current assets and liabilities. WCM helps the firm to meet its short-term obligations together with managing its assets. Firms can decrease their capital dependency and resource utilization for financial stability improvement through the management of working capital efficiently.

Working capital management is an integral part of functional financial management that aims to achieve its objective of maximizing shareholders' wealth and a firm's profitability. In particular, the investment and management decisions of different components of working capital like current assets and current liabilities to ensure positive feedback on a firm's profitability, liquidity, and shareholder value. There have been different views from the previous literature on the management of working capital and its importance in the financial performance of the organization. Few researchers (Baños-Caballero et al. 2016; Konak and Güner 2016; Zariyawati et al. 2017) found that investing or managing working capital benefits company to increase its sales, improve customer relations, decrease supply related costs, and others together with improving profitability. Whereas, other studies (Gonçalves et al. 2018; Ren et al. 2019) found negative impact of working capital on profitability as it leads to additional financial expenses, funds getting locked in working capital, and so on. Kieschnick et al. (2013) stated that focusing on working capital also leads the company to financial distress following to the bankruptcy probability. Thus, from the two different views, it is crucial to identify the optimal level of working capital that helps the company to improve their performance consistently. There is a need to balance or trade-off between the risk and profit. However, the trade-off between the profit and risk is not always confirmed or agreed by previous literature (B. Le 2019; Yoon and Jang 2005). In addition, economic conditions, and other company features are important to consider while maintaining the optimal level of working capital toward the performance.

Many previous researches have confirmed that WCM leads to the performance of the organization in both short and long term (Afrifa 2016; Altaf and Shah 2017; Baños-Caballero et al. 2014). Liquidity is heavily affected due to current assets and liabilities and thus, efficient WCM is the most important strategy for the organization to avoid any mis-management in excessive investments and monitoring current assets and liabilities for meeting short-term debt and other operational obligations. In the study performed by García-Teruel and Martínez-Solano (2007), investigating the role of WCM toward performance of Spanish SMEs showed that current assets hold around 69% of the total assets, whereas 52% of the current liabilities of the total liabilities. In addition, Van Horne and Wachowicz (2001) mentioned that managers give more attention to the management of current assets and current liabilities, if their portion in the total assets and total liabilities are higher. Less attention has been paid to the short-term debt or financial obligations by the firm and more focus has been given to long-term financial obligations. This lack of attention toward short-term working capital led many companies to financial decline or bankruptcy.

Working capital management is essential for manufacturing companies in Oman at the moment due to the unstable financial market conditions brought on primarily by uncertainties in macroeconomic fundamentals especially after the COVID-19 pandemic. One such phenomenon is pricing volatility, which necessitates careful working capital management on the part of business owners. This Oman study is significant and timely in light of the aforementioned as well as the recent financial sector clean-up, which has led to

the failure of numerous financial institutions. It provides useful facts and insight, especially to the listed industrial enterprises, on the need of effective working capital management in the pursuit of profitability.

Empirical research performed by Mandipa and Sibindi (2022) indicates that there is an ongoing debate on the relationship between working capital management and financial returns that increase shareholder value. By analyzing the relationship between working capital management and shareholder's wealth creation in the context of Oman, this study aims to close this knowledge gap and answer the research questions as: (1) How managing working capital would create shareholder's wealth? (2) what is the effect of earnings quality on shareholder's wealth? and, (3) Is there any relationship between sales growth and shareholder's wealth creation?

The remaining section for this paper is as follows: Section 2 provides the existing and relevant literature on the earnings quality–working capital management-shareholder's wealth relationship. Section 3 describes the empirical methods used for econometric analysis. Robustness checks and empirical analysis are provided in Section 4. Section 5 concludes the paper with policy implications.

## 2. Literature Review and Hypothesis Development

Maintaining liquidity is very crucial in order to meet the daily working capital commitments. Making profits and boosting shareholder wealth are the primary goals ensuring that the company is operating profitably and efficiently. There are inclinations for current assets and liabilities to be unequal, which impacts a company's profitability and growth (Amponsah-Kwatiah and Asiamah 2020). Many previous literatures (Hussain et al. 2021; Lefebvre 2022; Rey-Ares et al. 2021), on the relationship between WCM and performance in emerging and developed markets, have shown mixed results and majority of them found negative relationships. The firm's use of debt financing is anticipated to affect the value of a dollar invested for shareholders. In order to finance working capital, Lefebvre (2022) addressed the crucial significance that debt maturity decisions play. This is because the firm might not always have access to affordable short-term financing options. A mismatch between the maturity of the company's assets and liabilities, according to Preve and Sarria-Allende (2010), could result in suboptimal financing patterns and liquidity issues, which could lead to default and financial difficulties.

Empirical investigations on the aspects of working capital management linking with firm performance (Dechow 1994; Padachi 2006) and firm's profitability (Deloof 2003; García-Teruel and Martínez-Solano 2007; Marttonen et al. 2013; Sharma and Kumar 2011) and (Shankaraiah and Sudarshan 1986) for keeping balance between liquidity risk and profitability have been investigated. Raheman and Nasr (2007) analyzed the effect of working capital management on liquidity and profitability of the firm using a sample of 93 Pakistani firms and found that cash conversion cycle will lead to a decrease in profitability and positive shareholder value is created by reducing the cash conversion cycle at a possible minimum level. Likewise, Marttonen et al. (2013) analyzed the impact of working capital management on profitability studying industrial maintenance of service companies and found that due to light fixed assets and good profitability working capital management is over emphasized. Subsequently, Rafuse (1996) claimed that improving working capital by delaying payment to creditors and returns to shareholders will rarely produce any benefits to the organization. Shareholders will be benefited if the company reinvests the net profit or settles down the creditors rather than distributing it in the form of dividends (Panigrahi and Zainuddin 2015). Similarly, Abuzayed (2012) examined the effect of working capital management of firm performance and indicated that more profitable firms are less motivated to manage their net working capital. This study also revealed that . . . . "financial markets failed to penalize managers for inefficient working capital management" (Abuzayed 2012). However, the relationship between working capital management and shareholder's value has not been focused upon by previous academicians and practitioners especially for the airline industry that are struggling to provide safety, convenience, and financial sustain-

ability that concerns the industry future. The aforementioned studies have quantified the components of working capital management with a solid base describing the characters of gross and net working capital and estimate inter-relationship between working capital management and shareholder value through the use of a correlation model estimating the correlation coefficient (r).

According to Panigrahi et al. (2014), management decisions such as capital structure, dividend policies, remuneration, credit policy, and investment decision have great impact on shareholder's wealth maximization. Furthermore, Carney and Dostaler (2006) briefly outlined that the strategic and operational control of an airline is in the hand of managerial executives who serve as an agent for widely diversified shareholders. Consequently, when institutional investors control significant shares of voting equity, airline executives will tend to emphasize shareholder value maximization as a core strategic goal (Carney et al. 2011). Earlier studies like Silva (2011) and Karadagli (2012) found that there exists a relationship between working capital management and shareholder's return by reducing the number of days accounts payable to a minimum level. Thus, in corporate finance, the significance of WCM cannot be overstated due to its direct impact on the firm's liquidity and profitability (Amponsah-Kwatiah and Asiamah 2020).

The theoretical underpinnings of the research served as the basis for the study's framework, which include control variables such as sales growth and earning quality along with working capital management components such as days of working capital, cash conversion cycle, and payable deferred period. These elements and the controls were kept as independent variables, and the wealth and performance of the shareholders were kept as the dependent variable (proxied by EPS and ROA). The conceptual framework's core premise is that working capital management elements, firm-specific characteristics, and the shareholder wealth of businesses are related. This suggests that variations in any of the independent variables impact the shareholders' wealth. Figure 1 provides the relationship.

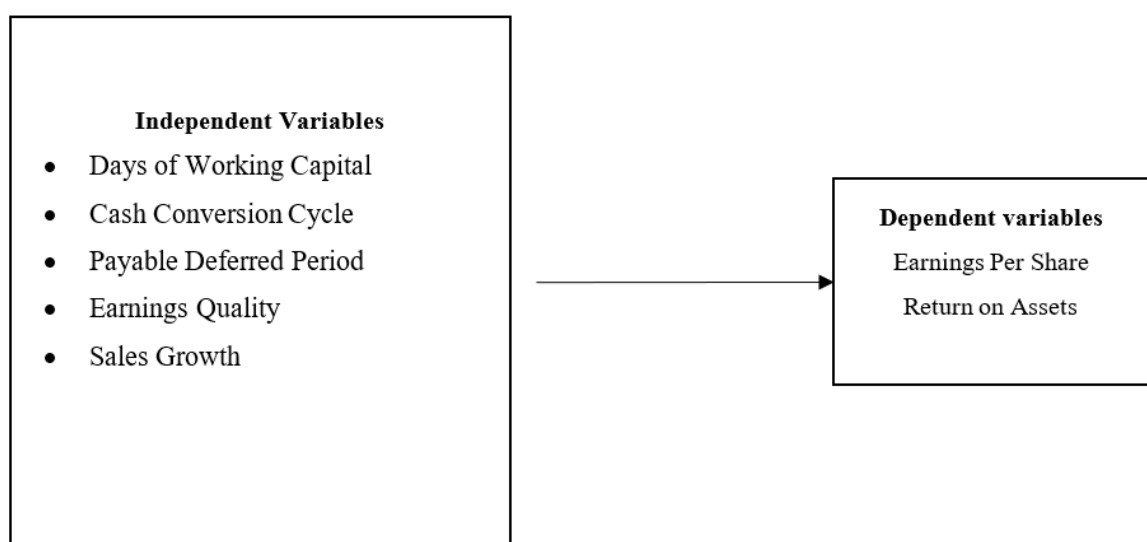

**Figure 1.** Conceptual model of the study. Source: constructed by authors.

The study's graphical flow is examined in this figure. DWC, CCC, PDP, EQ, and SG are used as the independent variables impacting on earnings per share and return on assets representing shareholder's wealth creation.

According to earlier studies, the management of WC makes a significant contribution to gross profit (Abuzayed 2012; Gul et al. 2013; Padachi 2006). In earlier research (Samiloglu and Akgün 2016; Seth et al. 2020a; Ukaegbu 2014), it was discovered that WCM and the gross benefit association frequently noticed the inverse link between these dimensions. The debt-to-equity ratio has been shown to have a detrimental influence on profitability (Öztürk and Karabulut 2018). On the basis of above empirical studies, we have utilized statistical

and econometric model for analyzing components of working capital management such as cash conversion cycle, and for analyzing shareholder value earnings per share was used.

From the previous literature, it can be argued that firms have paid more attention toward long-term financial decisions ignoring the short-term obligations. Thus, it must be an integral part of the business to provide attention on the short-term financial decisions and manage company's overall assets and liabilities. Thus, the main goal of this study is to empirically investigate the WCM-performance relationship for oil and gas companies in Oman. Next section provides evidence on literature review to support the hypotheses.

*2.1. Variables*

2.1.1. Working Capital Management and Shareholder's Wealth

Management of working capital is seen as a key factor in determining a company's profitability. Kusuma and Bachtiar (2018) define working capital management as the control of a company's current assets and liabilities. Working capital management consists of inventory control, accounts receivable, accounts payable, and the cash conversion cycle as its constituent parts. Businesses could have a working capital balance that maximizes their worth. Large inventories and a forgiving trade credit policy may, on the one hand, result in larger sales. A lower chance of stockout results from a larger inventory. Trade credit may increase sales since it enables consumers to evaluate product quality prior to making a purchase (Altaf and Shah 2018; Ren et al. 2019). Customers may find suppliers to be a more affordable source of credit as a result of the suppliers' potential cost advantages over financial institutions (Costa and Habib 2021). Giving out trade credit and maintaining inventories have the disadvantage of retaining money in working capital. More importantly, WCM is a vital force for an organization, and effective WCM is one of the prerequisites for a company's financial success (Amponsah-Kwatiah and Asiamah 2020). Studies have indicated that routines related to protecting cash and inventory as well as credit risk assessment are generally the ones that are carried out most frequently while managing working capital (Kabuye et al. 2019).

The aim to increase shareholder wealth, according to Sinnadurai et al. (2021), "drives our actions." Therefore, there is no question that the maximizing of shareholder wealth serves as the foundation for corporate finance. Numerous empirical studies in industrialized nations have focused on the implications of effective working capital management for shareholder value creation. The majority of researchers contend that effective working capital management is crucial to achieving the objective of maximizing shareholder wealth because it affects a firm's risk, profitability, and ultimately shareholders' wealth (Aktas et al. 2015; Boisjoly et al. 2020; Oseifuah and Gyekye 2017).

Therefore, it follows that any organization would benefit from effective working capital management. However, in some developed economies, including the USA, UK, Canada, and Australia, ineffective working capital management (WCM) has long been identified as a significant contributor to business failure (Hingurala Arachchi et al. 2017; Ramiah et al. 2016).

Following the global financial crisis of 2009, which severely taxed the financial resources of many businesses, it is not surprising that interest in WCM has grown more significant than ever before for businesses and researchers (Wang et al. 2020). Based on the arguments that have gone before, it is stated in this study that WCM is unquestionably a crucial component of the process of maximizing shareholders' wealth since effective WCM influences a firm's long-term survival as well as profitability, liquidity, and ultimately its value.

The firm's use of debt financing is anticipated to affect the value of an extra dollar invested for shareholders. In order to finance working capital, B. B. Le (2019) addresses the crucial significance that debt maturity decisions play. This is because the firm might not always have access to affordable short-term financing options. A mismatch between the maturity of the company's assets and liabilities, according to Preve and Sarria-Allende (2010), could result in suboptimal financing patterns and liquidity issues, which could lead

to default and financial difficulties. In conclusion, effective and efficient WCM guarantees that a firm will be able to continue its operations and that it has enough cash flow to cover both maturing short-term debt and impending operational expenses (Asare et al. 2022).

In recent years, research scholars have investigated the strategic role of working capital management in firms and indicated that WCM affects profitability (Kinasih Yekti Nastiti et al. 2019). According to Kwenda and Matanda (2015), maximizing shareholder wealth is dependent on effective working capital management. In both theoretical and empirical studies, working capital management has gained a lot of attention at the cost of capital structure and capital budgeting decisions. According to Tanveer et al. (2016), WCM is one of the controversial subjects in corporate finance. Because it affects return and profitability, it involves a number of financial decisions that are both challenging and essential to the success of any particular company. The fact that working capital management involves both current assets and current liabilities makes it extremely difficult. Rey-Ares et al. (2021) analyzed the relationship between WCM and profitability among 377 Spanish fish canning companies and found that profitability is related to the collection period and the inventory conversion period. Further, empirical evidence also revealed that increasing sales would benefit an optimal level of receivables. Alvarez et al. (2021) analyzed the relationship between WCM and the profitability of Argentine manufacturing firms and suggested that an increase in WCM determines an improvement in performance in terms of ROA and ROE. Conversely, leverage was negatively related to profitability.

Empirical evidence reveals that profitability is related to the collection period, therefore leading to an optimal level of receivables. Similarly, empirical data points to a convex or U-shaped relationship between inventory investment and financial profitability, i.e., that an initial increase in inventory levels will decrease the company's financial position until a point where the increase in inventory will result in an increase in financial profitability. Based on the previous arguments and empirical investigations, the following hypothesis has been formulated:

**H1.** *Working capital management significantly affects shareholder wealth creation.*

### 2.1.2. Earning's Quality and Shareholder's Wealth

The ability to manage working capital is closely related to earning quality. The percentage of income that can be directly linked to a company's main business operations is referred to as the quality of earnings. Many research (Abou-El-Sood and El-Sayed 2022; Alves 2021; Elzahaby 2021) illustrate the relationship between profit and investment decision-making, as well as the techniques and procedures utilized to ascertain how the quality of profit quality is generated by a firm. In order to illustrate the usefulness of decisions in specific decision contexts, academics have developed a number of "earnings quality" indicators over the years. Chen et al. (2022) measures the informativeness of country-level profits using earnings-based hedge portfolio returns and found that using accrual accounting more frequently than cash accounting is linked to lower profits informativeness only in nations with weak shareholder protection across 21 countries. Abdelghany (2005) using a sample of 90 NYSE listed companies suggested that before making any financial or investment decisions or taking any remedial action, the stakeholders must evaluate the quality of earnings using a variety of methods. Based on the previous arguments, second hypothesis is proposed as:

**H2.** *Earnings quality significantly affects shareholder's wealth.*

### 2.1.3. Sales Growth and Shareholder's Wealth

Many companies have valued sales growth and have included it in their yearly reports. The main factor in helping businesses improve their financial performance and profitability has been sales growth (Brush et al. 2000). However, academics have argued that growth occasionally serves managers rather than stockholders. Groza et al. (2021) proposed that sales intellectual stimulation helps to drive sales growth and organizational innovativeness.

The results indicated that the relationship between organizational innovativeness and sales growth follows a non-linear U-shape. The perceptions of senior managers are significantly influenced by sales growth targets. Peesker et al. (2021) found through questionnaires that the most frequently indicated goal by senior managers was sales. According to Argenti (2018), planning systems often start with sales targets. Emphasizing sales growth also offers management a practical and obvious standard to strive for. We are aware of very little research that empirically explores the relationship between sales growth and firm performance, despite the significant role that sales growth plays in the corporate world and its fundamental place in agency theory. This research offers the first attempt to address this problem by examining how sales growth affects shareholder wealth. Based on these arguments, we proposed our third hypothesis:

**H3.** *Sales growth significantly affects shareholder wealth creation.*

### 3. Methodology

The study's methodology is presented in this section. It includes descriptions of the research design, model specification, definition, and measurement of model variables, estimating methodologies, data sources, and data analysis tools. The link between WCM, its discrete components, and shareholder's wealth creation was estimated using the model below. This is how the model can be expressed:

*3.1. Sample and Data*

The study sample included 31 stock market listed companies employed in the manufacturing sectors using annual data from 2004 to 2019 with 350 observations in total. Data was extracted from the annual reports of the listed manufacturing companies in MSE. The annual data for 2020 and 2021 were not available in the company's website or the MSE. The selection of the sample was based on the availability of the data, their importance, and contribution toward the economy improvement of the country.

Deductive methodology with an explanatory research design was used in this study. The objective of this approach is to establish a causal connection between the variables (Sanders et al. 2016). The deductive method involves analyzing a theory by formulating specific hypotheses and acquiring information to confirm or refute them (Bryman and Cramer 2012). Within the scope of classical and neoclassical economics, this adheres to the positivist paradigm.

*3.2. Model Estimation*

The following static panel models were used after reviewing the views of (Agyemang Badu and Asiedu 2013; Amponsah-Kwatiah and Asiamah 2020; Hussain et al. 2021). These models were used in order to illustrate the importance of the variations among the firms and the precise effects of the selected variables inside the firms over the timeframe.

Fixed effect models

$$ROA_{it} = a_i + \beta_0 + \beta 1DWC_{it} + \beta 1DWC_{it} + \beta 2CCC_{it} + \beta 3PDP_{it} + \beta 4SG_{it} + \beta 5EQ_{it} + \varepsilon_{it} \quad (1)$$

$$EPS_{it} = a_i + \beta_0 + \beta 1DWC_{it} + \beta 1DWC_{it} + \beta 2CCC_{it} + \beta 3PDP_{it} + \beta 4SG_{it} + \beta 5EQ_{it} + \varepsilon_{it} \quad (2)$$

$$ROA_{it} = \beta_0 + \beta 1DWC_{it} + \beta 1DWC_{it} + \beta 2CCC_{it} + \beta 3PDP_{it} + \beta 4SG_{it} + \beta 5EQ_{it} + u_{it} + \varepsilon_{it} \quad (3)$$

$$EPS_{it} = \beta_0 + \beta 1DWC_{it} + \beta 1DWC_{it} + \beta 2CCC_{it} + \beta 3PDP_{it} + \beta 4SG_{it} + \beta 5EQ_{it} + u_{it} + \varepsilon_{it} \quad (4)$$

To determine whether a random or fixed effect model is acceptable, the Hausman test has been employed frequently. The following is stated as the Hausman test hypothesis:

**H0.** *Random effect is the preferred model.*

**H1.** *Fixed effect is the preferred model.*

The findings of the Hausman test show that the alternative hypothesis is accepted in favor of the null hypothesis when the *p*-value is less than 0.05.

Where, $\alpha$ = Constant (the intercept, or point where the line cuts the Y axis when X = 0)

$\beta_0$ = 5 Constant (the intercept, or point where the line cuts the Y axis when X = 0)

$\alpha_i$ = Firm specific effect variable

$\varepsilon_{it}$ = within firm error

$u_{it}$ = Between firm error

In order to analyze the data, Eviews 12 was used as an econometric software.

### 3.3. Variables

For this analysis, return on assets (ROA) and earnings per share (EPS) was used as a dependent variable representing shareholder's value. The independent variables include the components related to the management of working capital as shown in Table 1. The independent variables considered related to the management of working capital include days in working capital (DWC), cash conversion cycle (CCC), payable deferred period (PDP), sales growth (SG) and earnings quality (EQ). The series on ROA and EPS (proxies for shareholder's wealth) were sourced from the annual reports of the listed manufacturing firms on the Muscat stock exchange website, whilst the series on days in working capital (DWC), cash conversion cycle (CCC), payable deferred period (PDP), sales growth (SG), and earnings quality (EQ) were sourced from the individual firm's annual reports.

**Table 1.** Working capital management measurements.

| Variables | Measurement |
|---|---|
| Days in Working Capital (DWC) | Receivable + Inventory − payable/(sales/365) |
| Cash conversion cycle (CCC) | DIO + DSO − DPO |
| Payable deferred period (PDP) | accounts payable/COGS × 365 |
| Sales Growth (SG) | Difference in sales at time t |
| Earnings quality (EQ) | Net cash from operating activities divided by net income |

Note: WCM—working capital management; COGS—cost of goods sold; DIO—days inventory outstanding; DSO—days sales outstanding; DPO—days payable outstanding.

In this research, we used days in working capital (DWC) which was measured by adding accounts receivables, and inventory and subtracting accounts payable. DWC describes the days taken by the company to convert its working capital into revenue. Companies that take less time are more efficient than companies that take a long time to convert their working capital into revenues. The next component of WCM was cash conversion cycle (CCC) which expresses the time taken to convert its inventory investments into cash. CCC is one of the quantitative measures that help to evaluate the efficiency of the company and its operations. CCC will differ by the sector and the nature of the business. Payable deferred period (PDP) was the next WCM component considered for the research. PDP is the time that the company takes to do payments to its suppliers or creditors. In order to avoid delays in receiving payments, suppliers give discounts to the firms that pay faster. Companies that pay back to their creditors and suppliers faster are more efficient. Sales growth (SG) was the next variable that is measured as changes in revenue over a fixed period of time. Firms that show high growth rates are efficient enough to capitalize on their finance activities (Eka 2018). Several studies have used sales growth to investigate its effect on profitability (Kinasih Yekti Nastiti et al. 2019; B. Le 2019; Nguyen et al. 2020; Tarkom 2022) and financial performance (Kabuye et al. 2019; Kinuthia et al. 2019; H. Le et al. 2018). Earnings quality (EQ) measures the reliability of the company in assessing its current and future performance. Dechow et al. (2010) reviewed proxies for earnings quality like accruals, smoothness, timeliness, loss avoidance, investor responsiveness, and security exchange enforcements to improve the firm's fundamental performance. The results found earnings quality to be an important factor for the efficiency and effectiveness of the firms.

*3.4. Data Analysis*

Analyses that were descriptive and quantitative of the investigation were used. To help with the descriptive and inferential analysis, tables were presented. Using the E-views 12.0 software package, the data were processed, and descriptive statistics, correlation, and panel regression were used to analyze the data. Once more, the least squared dummy variable (LSDV) estimator was used to estimate the fixed effect model, while the generalized least squares estimator was used to estimate the random effect model (GLS). Due to the nature of the relevant variables and the fact that these estimators are suitable for reliable estimations, they were used.

## 4. Results and Discussions

The results and discussion of the study are presented in this section. Prior to discussing the findings, correlation analysis, static panel regression analysis, and descriptive statistics are presented.

Table 2 displays the descriptive statistics for the pertinent variables. The average values of a bunch of variables were calculated using the mean. The standard deviation calculated the degree to which the values deviated from the mean. The range of the variables was encapsulated by the minimum and maximum values. There were 350 observations in all. Table 2 demonstrates that all of the variables had average values that were positive (means). There was significant fluctuation around the averages for the dependent variables ROA and EPS, which had means of ($-0.33$; 0.3) that were less than their standard deviations (458.7; 0.7). Jarque Bera results were performed to detect the normality of the data. The value of Jarque Bera was too large indicating that the errors are not normally distributed due to non-parametric data.

**Table 2.** Descriptive statistics for the variables.

|  | **ROA** | **EPS** | **DWC** | **CCC** | **PDP** | **SG** | **EQ** |
|---|---|---|---|---|---|---|---|
| Mean | $-33.0$ | 0.3 | 124.5 | 146.1 | 72.0 | 0.1 | 5.8 |
| Median | 5.2 | 0.1 | 132.2 | 156.9 | 44.8 | 0.1 | 1.0 |
| Maximum | 166.7 | 2.4 | 2190.0 | 1368.8 | 730.0 | 3.2 | 78.3 |
| Minimum | $-8000.0$ | $-6.6$ | $-12,410.0$ | $-16,911.7$ | 0.0 | $-1.0$ | $-128.6$ |
| Std. Dev. | 458.7 | 0.7 | 709.1 | 951.5 | 97.6 | 0.3 | 16.6 |
| Skewness | $-16.1$ | $-2.5$ | $-16.4$ | $-17.2$ | 3.2 | 3.5 | 0.4 |
| Kurtosis | 274.0 | 36.2 | 291.7 | 308.4 | 16.8 | 30.3 | 20.0 |
| Jarque-Bera | 1,045,665.0 | 15,861.6 | 1,185,332.0 | 1,326,101.0 | 3275.3 | 11,110.3 | 4042.6 |
| Probability | 0.0 | 0.0 | 0.0 | 0.0 | 0.0 | 0.0 | 0.0 |
| Observations | 352 | 352 | 352 | 352 | 352 | 352 | 352 |

Sources: Constructed by authors using Eviews.

*4.1. Correlation Results*

Table 3 displays the pair-wise correlation between the various variables. Positive and significant association between ROA and DWC was found (0.329). This indicates that these two focal variables are linked and move together. Additionally, there was a positive and substantial association between ROA and SG, EQ and CCC (0.217, 0.062 and 0.91). Additionally, since these focus factors were linked, it follows that ROA improves as SG, EQ, and CCC do.

As a result, these factors are crucial to our investigation. Furthermore, there was a positive and significant association between ROA and EPS (0.082). This indicates that the focus factors were likewise correlated, suggesting that when EPS rises, ROA will follow. Payable deferred period (PDP) and the measure of ROA had negative and significant correlations, respectively, with coefficients of 0.134.

Additionally, with values of 0.006 and 0.317, the association between DWC and PDP and the measures of EPS were negative. The purpose of the correlation analysis was to demonstrate the strength of the relationship between the variables employed in the analysis

and to avoid collinearity. The regression analysis revealed the explanatory variables' effects on ROA and EPS.

**Table 3.** Correlation statistics for the variables.

|  | **ROA** | **EPS** | **DWC** | **CCC** | **PDP** | **SG** | **EQ** |
|---|---|---|---|---|---|---|---|
| ROA | 1 |  |  |  |  |  |  |
| EPS | 0.082556 | 1 |  |  |  |  |  |
| DWC | 0.869614 | −0.00682 | 1 |  |  |  |  |
| CCC | 0.910566 | 0.006453 | 0.990722 | 1 |  |  |  |
| PDP | −0.13697 | −0.31787 | −0.08396 | −0.09271 | 1 |  |  |
| SG | 0.217708 | 0.053505 | 0.119457 | 0.140546 | −0.11658 | 1 |  |
| EQ | 0.062931 | 0.796067 | −0.01855 | −0.00777 | −0.24848 | 0.038344 | 1 |

Sources: Constructed by authors using Eviews.

### 4.2. Effect of WCM on ROA

As shown in Table 4, the results of the fixed effects model, which were reported in this section, met the goals of the study with regard to how working capital management components affected ROA. Results in Table 4 show that the model can account for 83 percent of the fluctuations in ROA with an adjusted R square of 83 percent. Only results with random effects are analyzed using the Hausman test statistic of (0.001).

**Table 4.** Effect of WCM components on ROA.

| **Variable** | **FE** | **RE** |
|---|---|---|
| Days in Working Capital | −1.091 *** | −1.44 *** |
| Cash Conversion Cycle | 1.160 *** | 1.52 *** |
| Payable Deferred Period | 0.060 ** | 0.08 |
| Sales Growth | 9.200 * | 10.26 |
| Earnings Quality | 0.699 *** | 0.089 * |
| Constant | −75.39 *** | −92.84 *** |
| N | 352 | 352.0 |
| No of companies | 31 | |
| Hausman test (Prob > χ2) | 0.001 | |
| R-squared | 0.83 | 0.94 |

**Notes**: t-statistics in parentheses * $p < 0.1$, ** $p < 0.05$, *** $p < 0.01$; FE—fixed effect, RE—random effect; Sources: Constructed by authors using Eviews.

As a result, findings revealed that the cash conversion cycle (CCC) coefficient was favorable and statistically significant at the 1% level of significance, with a coefficient of 1.160 units. Accordingly, a unit increase in the firms' effective cash conversion cycle will, all other things being equal, result in a 1.160 unit rise in the firms' ROAs. This suggests that the cash conversion cycle has a favorable impact on ROA.

Similarly, it was also found that earnings quality (EQ) coefficient was favorable and statistically significant at the 1% level of significance, with a coefficient of 0.699 units. Accordingly, a unit increase in the firms' earnings quality will, all other things being equal, result in 0.699 unit rise in the firms' ROAs. This suggests that the earnings quality has a favorable impact on ROA.

The findings typically show that manufacturing companies' financial performance improves when working capital is adequately managed. The outcome is in line with the research of Ukaegbu (2014), Baños-Caballero et al. (2012), Wang et al. (2020) and Ding et al. (2013), all of whom claim that a company's profitability benefits from proper working capital management. Panda and Nanda (2018) also looked into the relationship between working capital management and firm profitability (ROA) in light of the expansion possibilities for manufacturing companies in Accra. The cash conversion cycle theory is also shown by this. This disproves the null hypothesis since it suggests that a firm's inventory management may affect its profitability.

The findings further demonstrate that at a 1% level of statistical significance, the coefficient of days in working capital is negatively and statistically significant with a coefficient of −1.091 units. This indicates that all other things being equal, a unit increase in the days of working capital (DWC) will result in a 1.091 unit fall in the firms' ROAs. This suggests that working capital days have a negative impact on ROA. This further implies that manufacturing companies' financial performance is improved by increasing the number of working capital days as part of their working capital management strategy. The results confirm the findings of Akey (2019) who found a negative effect of days in working capital on profitability. It also contradicts that of Mohamad and Saad (2010) and Akinlo (2012). The findings of Preve and Sarria-Allende (2010) and Lefebvre (2022), which claimed that an appropriate and effective account receivable was favorable for a firm's profitability, are in conflict with the results.

### 4.3. Effect of WCM on EPS

This section summarizes the findings from the fixed effects model, which met the goals of the investigation into how working capital management affects profits per share (EPS), as shown in Table 5.

**Table 5.** Effect of WCM components on EPS.

| Variable | FE | RE |
|---|---|---|
| Days in Working Capital | −0.000 | −0.000 |
| Cash Conversion Cycle | 0.000 | 0.000 |
| Payable Deferred Period | −0.000 | 0.000 |
| Sales Growth | 0.022 ** | 0.023 ** |
| Earnings Quality | 0.037 *** | 0.037 *** |
| Constant | 0.08 *** | 0.087 |
| N | 352 | 352.0 |
| No of companies | 31 | |
| Hausman test (Prob > $\chi^2$) | 0.437 | |
| R-squared | 0.88 | 0.74 |

Notes: t-statistics in parentheses ** $p < 0.05$, *** $p < 0.01$; FE—fixed effect, RE—random effect; Sources: Constructed by authors using Eviews 12.

According to Table 5, the model accounted for 74% of the variability in EPS with an adjusted R2 of 74%. Only results with fixed effects are analyzed, once more using the Hausman test statistic of (0.437). As a result, findings of Table 5 demonstrate that the correlation between working capital days, cash conversion cycle, and payable deferred time was negative and not statistically significant at any level of significance. This indicates that elements of working capital management were powerless to affect EPS-based measures of shareholder wealth. This suggests that investors are unconcerned with how businesses handle their working capital.

Findings of Table 5 further demonstrate that the sales growth and earnings quality coefficients are positive and statistically significant at the 5% level of significance with coefficients of 0.022 and 0.037 units, respectively. This means that, if all other factors are equal, a unit rise in the firms' sales growth and earnings quality will result in an increase in their respective EPS of 0.022 and 0.037 units. This suggests that EPS is positively impacted by both sales growth and earnings quality. This shows that boosting manufacturing companies' sales growth and earnings quality results in more wealth creation for their shareholders.

### 5. Conclusions

This study aimed to investigate how working capital management affected the wealth of shareholders in manufacturing listed companies in Oman. A panel approach was used inside an explanatory research design to accomplish the study's goals. The descriptive statistics of the variables were used as the first step in the paper's analysis of the data. Additionally, a correlation study was carried out, which revealed both positive and negative

connections between the explanatory variables and the ROA and EPS, which served as stand-ins for shareholder wealth.

The study discovered a strong positive and significant association between the cash conversion cycle, payable deferred period, sales growth, earnings quality, and ROA based on the fixed and random effect results based on the Hausman test. The association between days in working capital, cash conversion cycle, and payable deferred period and ROA was negative and favorable. This negative result implies that reducing the days in working capital, cash conversion cycle, and payable deferred period r would increase the return on assets. However, the overall results show that the Omani manufacturing sector's ROA is positively influenced by the explanatory variables sales growth and earnings quality.

There are numerous theoretical, sociological, and practical ramifications of the study.

The results of this study have been able to support the literature regarding the impact of WCM on the wealth of shareholders in manufacturing firms. The results have thus established a foundation for further research, improving future scholars' comprehension of the connection between WCM and shareholder wealth of manufacturing firms.

Firms must recognize the importance of seen working capital management as forming a holistic decision-making and control system that acts as a driver of financial performance for companies of different industries and sizes it represents the link between liquidity and profitability (Lyngstadaas 2020) and (Akgün and Karataş 2020). Therefore, sound working capital management can be a competitive advantage for the firm as strict working capital measures can positively affect the market values of the firms and boost their performance metrics (Boisjoly et al. 2020). Working capital management plays a vital role in the addition of value, generation of profitability and the risk assessment of any firm. Especially since working capital measures composes the bases for making short-term investment and financing decisions (Seth et al. 2020a) and if a company is unable to manage their working capital it would lead to business failure (Mabandla and Makoni 2019).

Effective working capital management is even more imperative for some sectors like the manufacturing sector since they play a major role in economic development. Domestic manufacturing firms should be extra cautious with managing their current assets and current liabilities to keep pace with global competition and thrive (Khan et al. 2019). A constant monitoring of working capital measures is even more crucial in developing and emerging economies as listed firms in less developed economies fail to achieve optimal efficiency of their working capital programs Alvarez et al. (2021).

The study's practical conclusions have the following effects on manufacturing companies: (1) Effective stock, creditors, debtors, and cash flow management are all crucial for improving a company's profitability. (2) Policymakers will be able to administer performance standards and expertise on managing working capital for manufacturing firms. (3) Management teams of manufacturing firms will be able to determine the optimized inventory level and accounts receivable level which will be beneficial for their managing inventory and receivable control.

Based on the findings, the paper implies that working capital management variables including inventory management, account receivables, account payables, and cash conversion cycle be highlighted in management policies for manufacturing firms because they are crucial to the shareholders wealth of the company. Shareholders should select companies that not only have higher profitability but also fully reveal their sustainability reports when making judgments.

The study has some limitations, despite being able to demonstrate clearly how WCM affects shareholder wealth of listed manufacturing enterprises. The biggest drawback is the lack of data for several industrial companies in regard to some important variables taken into account in the study. Only 31 manufacturing companies were thus covered. The annual data for 2020 and 2021 of many companies were not available in their website or the MSE. However, the study's findings provide information that will be helpful to all manufacturing companies. In order to maintain consistency, future studies might take into account examining more of these manufacturing companies if data are available. Internal

control systems are important when looking at how working capital management affects the profitability of manufacturing companies.

**Author Contributions:** S.K.P. and M.W.A.K. formulated the study design. S.K.P., M.W.A.K. and M.J.A.F. designed and conceived the research methodology. S.K. and F.R. collected and formulated the data as per the software requirement. S.K.P. and S.K. analyzed and interpreted the data. S.K.P., S.K. and F.R. finalized the paper. All authors have read and agreed to the published version of the manuscript.

**Funding:** The authors would like to acknowledge the financial support by University of Buraimi for this research work.

**Informed Consent Statement:** Not applicable.

**Data Availability Statement:** The dataset generated during the study can be accessed using the following URL: panigrahi, shrikant (2022), "WCM DATASET", Mendeley Data, V1, doi:10.17632/k9sg2t2nz7.1.

**Acknowledgments:** The authors highly acknowledge the university of Buraimi, Oman for their financial and technical support for completing this research work.

**Conflicts of Interest:** The authors declare no conflict of interest.

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
