# Peer review of "Working Capital Management and Shareholder’s Wealth Creation: Evidence from Manufacturing Companies Listed in Oman"

_ijfs, doi:10.3390/ijfs10040089_

Round 1

Reviewer 1 Report

The authors should consider the following recommendations in order to improve the original manuscript:

- The abstract should be more consistent with the main text of the paper, preferably structured, simple, specific, clear and unbiased.

- The keywords should not overlap with the title of this paper.

- To include certain relevant research questions.

- Authors also did not provide sufficient evidence on literature review to support the hypotheses.  Authors should take into consideration much more recent publications in the sphere of discussed subject matter, especially studies conducted during the last 5 years.

- Authors argued that “The study uses a balance panel of 31  manufacturing listed firms from 2004 to 2019”. Why this panel of only 31  manufacturing? It is relevant and statistically representative? Why the time interval from 2004 to 2019 since we are in mid August 2022?

- Please discuss about Covid-19 pandemic caused by Severe Acute Respiratory Syndrome Coronavirus 2 (SARS-CoV-2) and its impact on economy. I suggest extending the literature section by including recent and relevant studies, such as for instance:

1.      Pourmansouri R, Mehdiabadi A, Shahabi V, Spulbar C, Birau R. An Investigation of the Link between Major Shareholders’ Behavior and Corporate Governance Performance before and after the COVID-19 Pandemic: A Case Study of the Companies Listed on the Iranian Stock Market. Journal of Risk and Financial Management. 2022; 15(5):208. https://doi.org/10.3390/jrfm15050208.

- Human proofreading, English grammar and spelling correction are also required in order to improve the quality of the manuscript.

Reviewer 2 Report

The title is so long and not clear for readers that ignore what do you mean by "working capital management", "earnings quality" or "shareholders´ wealth creation". In fact, apparently, it is difficult to put together these three constructs in the same sentence.

Please, avoid acronyms in the abstract. If you decide to use acronyms, do it in extense (such as you do it in ROA) and not as EPS, that has no explanation. 

Introduction: You must clearly define from the beginning concepts such as "earnings quality" and "working capital". Instead of this, you start linking both constructs without a minimum explanation of the nature of your research. Readers will not understand your research if you do not explain what is to be studied/analyzed, the logic of your concerns, why it is relevant/important, and how you will proceed in the paper. Thus, first, define the concepts, second, you must explain the expected relationship based on a relevant literature review defining the objective of the research and its soundness, third, show the structure of the sections of the paper as you have done. To sum up, it is a question of reorganizing your introduction by changing the first strong sentences to a second level, giving the first place to define concepts and contextualize the study focusing on the soundness of the study.

In addition, a brief explanation of the listed companies system in Oman should be interesting, for readers without previous knowledge of the country. The question is, is this study useful/interesting in/for other contexts/countries? You must reflect on this point from the beginning. 

Theoretical background: In my opinion, nowadays, any researcher can avoid the Theory of Stakeholders, even when the support of the paper is the Theory of Shareholders. Please, reconsider your theoretical background to acknowledge (and include as a necessary complement if agree) that any business has to get financial, but also economic, social, and environmental goals. Otherwise, your paper will be out of the theoretical general framework of sustainability in business, even when considering financial issues, and even in this case, you should explain very well why you adopt an updated framework.

Variables and Results: All variables must be defined/explained. My suggestion is to expand point 3.3 from line 293 to line 303 to carefully explain each variable. Alternatively, you can develop table 1, including all variables and explanations. By the way, all tables or figures must be indicated and explained in the text. For instance, in table 2, I wonder 1) what is Jarque-Vera and 2) why it is relevant reporting this result on the table but 3) any comment in the text. Please explain every relevant result and avoid information that is not needed or relevant for your study.

Negative results or unexpected results must be very well explained and interpreted. Lines 430 to 434 deserve a credible explanation. Why the expected association was unfavorable?

Conclusion: Attention in lines 435 to 439 because you include theories that have not been previously considered. The conclusion is not for adding information. Be careful at this point. I agree with your comment about these theories but to write that in conclusion, you must improve very much your previous theoretical section.

Minor changes:

- So many errors in citing your references, for instance, line 106, Juan is a name, the surname is García-Teruel, so "Juan" must be deleted from the citation.

I hope my review serves for improving your paper. Good luck

Round 2

Reviewer 1 Report

The original manuscript has been improved. The authors followed the recommendations included in the previous review report so that the quality of their research article has increased. I also appreciate the effort of the authors in this regards.

Reviewer 2 Report

Thanks for taking into consideration my recommendations. Your article is better now and suitable for publication however I continue thinking that the title is so long. In addition, in your abstract EPS remains when the correct will be "earnings per share (EPS)". All the best.